# Rituximab-Containing Treatment Regimens May Imply a Long-Term Risk for Difficult-To-Treat Chronic Hepatitis E

**DOI:** 10.3390/ijerph17010341

**Published:** 2020-01-03

**Authors:** Marten Schulz, Paula Biedermann, Claus-Thomas Bock, Jörg Hofmann, Mira Choi, Frank Tacke, Leif Gunnar Hanitsch, Tobias Mueller

**Affiliations:** 1Department of Hepatology and Gastroenterology, Charité—Universitätsmedizin Berlin, CVK, 13353 Berlin, Germany; frank.tacke@charite.de (F.T.); tobias.mueller@charite.de (T.M.); 2Division of Viral Gastroenteritis, Hepatitis Pathogens and Enteroviruses, Department of Infectious Diseases, Robert Koch Institute, 13353 Berlin, Germany; BiedermannP@rki.de (P.B.); BockC@rki.de (C.-T.B.); 3Institute of Virology, Charité Universitätsmedizin Berlin, Labor Berlin—Charité-Vivantes GmbH, 13353 Berlin, Germany; joerg.hofmann@laborberlin.com; 4Department of Nephrology and Intensive Care Medicine, Charité Universitätsmedizin Berlin, 13353 Berlin, Germany; mira.choi@charite.de; 5Institute of Medical Immunology, Charité—Universitätsmedizin Berlin, 13353 Berlin, Germany; leif-gunnar.hanitsch@charite.de

**Keywords:** hepatitis E, rituximab, ribavirin resistance, hypogammaglobulinemia: CD4+ T cell lymphopenia

## Abstract

Hepatitis E virus (HEV) infection is an emerging disease in industrialized countries which is usually characterized by a self-limited course. However, there is an increased risk of HEV persistence in immunocompromised risk populations, comprising patients following solid organ transplantation or hematological malignancies. Recently, chronic HEV infection following rituximab-containing treatment regimens has been described. Here we report five patients with chronic hepatitis E after prior rituximab therapy for various indications. We determined the immunological characteristics of these patients and analyzed the development of ribavirin (RBV) treatment failure-associated mutations in the HEV genome. One patient became chronically HEV-infected 110 months after administration of rituximab (RTX). Immunological characterization revealed that all patients exhibited significant hypogammaglobulinemia and CD4+ T cell lymphopenia. One patient permanently cleared HEV following weight-based ribavirin treatment while three patients failed to reach a sustained virological response. In depth mutational analysis confirmed the presence of specific mutations associated with RBV treatment failure in these patients. Our cases indicate that rituximab-containing treatment regimens might imply a relevant risk for persistent HEV infection even years after the last rituximab application. Moreover, we provide further evidence to prior observations suggesting that chronically HEV infected patients following RTX-containing treatment regimens might be difficult to treat.

## 1. Introduction

Hepatitis E virus (HEV) infection is an emerging disease in industrialized countries which is usually asymptomatic and self-limiting [1]. Patients who fail to clear HEV three months after the onset of infection are defined as persistently infected, and weight-based ribavirin (RBV) treatment is recommended as standard first-line therapy in these patients [2]. Effective second-line treatment options in patients who fail to achieve sustained virological response (SVR) following RBV first-line therapy appear to be limited and comprise prolonged RBV retreatment periods or pegylated interferon. We and others have demonstrated that the potent hepatitis C polymerase inhibitor sofosbuvir, which showed anti-HEV effects in vitro [3], appears to be an ineffective rescue therapy in these difficult-to-treat patients [4]. Moreover, these patients are at risk of an aggressive course of their chronic HEV infection with rapid progression to end-stage liver cirrhosis [5].

There is an increased risk of HEV persistence in patient populations under immunosuppression after solid organ transplantation receiving calcineurin inhibitors, corticosteroids, mycophenolic acid, cyclosporine A, or mammalian target of rapamycin (mTOR) inhibitors, often administered as combined drug therapy [6,7,8], and in patients with hematological malignancies [9]. Persistent hepatitis E has also been described in patients with idiopathic CD4+ T lymphopenia and human immunodeficiency virus (HIV) infection with low CD4+ T-cell counts [10,11]. Diagnosis of chronic HEV infection in these risk populations is usually based on PCR since the humoral response resulting in anti-HEV antibodies immunoglobulin M/G (IgM/IgG) is often delayed or even absent. Anecdotal cases of persistent HEV infection during or following different treatment regimens involving rituximab (RTX) have been described recently with mixed response rates to standard RBV or interferon treatment as depicted in Appendix A [9,12,13,14,15,16,17]. RTX is a monoclonal CD20 antibody that leads to B-cell depletion for an extended period. Beside hepatotoxicity, reactivation of viral infections, especially hepatitis B virus, is a relevant and potentially life-threatening side effect of RTX-containing treatment regimens. Therefore, current hepatitis B clinical practice guidelines recommend hepatitis B testing in all patients undergoing RTX treatment to avoid fulminant hepatitis B reactivation [18].

Here we examined five patients with chronic hepatitis E following prior RTX therapy for various underlying diseases. We also determined the immunological characteristics of these patients and analyzed the development of previously described mutations in the HEV genome, which might affect the outcome of ribavirin treatment [19,20,21,22,23,24,25].

## 2. Materials and Methods

Mutational Analyses of the HEV Polymerase region.

Viral RNA was extracted from EDTA-plasma using the QIAamp Viral RNA mini Kit (Qiagen, Hilden, Germany) and the QIAcube (Qiagen, Hilden, Germany) according to the manufacturer’s instructions. RNA was reverse transcribed into cDNA with SuperScript IV Reverse Transcriptase (Invitrogen, Carlsbad, CA, USA) per manufacturer’s recommendations except with a prolonged incubation step of 20 min at 50 °C and 15 min at 55 °C. Semi-nested PCR for amplification of the RNA-dependent RNA-polymerase (*RdRp*)-gene of the HEV genome was performed using the Phusion Hot Start II High-Fidelity DNA Polymerase (Thermo Fisher Scientific, Waltham, MA, USA). Sense primer HEV-247_f and antisense primer HEV-128_r were used for the first PCR. Sense primer HEV-247_f and antisense primer HEV-248_r were used for the second PCR (Table 1). PCR conditions were: 30 s at 98 °C and 35 cycles consisting of 5 s at 98 °C, 30 s at 61 °C, 1 min at 72 °C, followed by 2 min at 72 °C. Amplicons were sequenced using the primers HEV-27, HEV-38, HEV-39, HEV-165, HEV-166, and HEV-247 with the BigDye Terminator (version 3.1) cycle sequencing kit (Applied Biosystems, Waltham, MA, USA) and the Applied Biosystems 3500 Dx Series Genetic Analyzer (Thermo Fisher Scientific, Waltham, MA, USA) (Table 1).

Resulting sequences were analyzed using the software Geneious 11.1.5 (Biomatters, Auckland, New Zealand). The sequences were analyzed regarding the previously reported seven mutations in the HEV-Polymerase region possibly associated with RBV treatment failure [19,20,21,22,23,24,25]. For genotyping the HEV Genotyping Tool https://www.rivm.nl/mpf/typingtool/hev/ (RIVM—Netherlands National Institute for Public Health and the Environment, Bilthoven, Netherlands) was used [23].

## 3. Results

### 3.1. Patients

#### 3.1.1. Case 1

A 69-year old male received seven cycles of R-CHOP plus two cycles of RTX in 2008 and 2009 for non-Hodgkin’s, lymphoma leading to long-term remission. A regular follow-up examination in July 2016 revealed elevated aminotransferases indicating hepatitis. An extended laboratory examination including HEV serology without HEV RNA testing failed to unravel the underlying cause. A liver biopsy obtained in September 2016 exhibited histological features of viral hepatitis, but laboratory results remained negative for hepatotropic viruses. The patient remained undiagnosed until HEV RNA quantitation was performed in March 2018, revealing a HEV load of 4.15 E6 copies/mL despite persistent HEV seronegativity until present (recomWell HEV IgG/IgM and for confirmation recomLine HEV IgG/IgM, both Mikrogen Neuried, Germany). Weight-based RBV was administered at a dose of 1000 mg/day for three months and the patient cleared HEV, which led to the termination of therapy. HEV relapsed after 4 weeks and RBV treatment was re-induced at a dose of 1000 mg/d for 4 additional months without achieving viral clearance. After a treatment pause of two months RBV was continued at a reduced dose of 800 mg/d due to increasing fatigue and insomnia of the patient, who shows persisting HEV replication and constantly elevated aminotransferases until present.

#### 3.1.2. Case 2

A 64-year old male with elevated aminotransferases presented himself at our outpatient clinic. He had been diagnosed with a monoclonal gammopathy of undetermined significance in 1992. In 2012, he had received eight cycles of bendamustine and RTX for Waldenstrom macroglobulinemia. Due to a relapse in 2016 he had been retreated with six cycles of bendamustine and RTX until February 2017, with stable remission since then. In November 2017, elevated aminotransferases were noted for the first time and the patient reported a short period of jaundice and fatigue. Extended laboratory testing revealed a HEV load of 6.1 E7 copies/mL. HEV IgG was positive and HEV IgM negative. After three months of persistent HEV replication, weight-based RBV treatment was started at a dose of 800 mg/d leading to virus clearance after three months of RBV treatment. RBV was stopped, but HEV relapsed only one month later and the patient was retreated with 800 mg RBV/d for an additional three months, achieving a sustained virological response.

#### 3.1.3. Case 3

A 22-year old male patient had been treated with eculizumab from 2013 to 2015 for two years followed by RTX due to an atypical hemolytic-uremic syndrome and immune thrombocytopenia. RTX was administered from May to June 2015 (three cycles). In the following years, the patient suffered from recurrent infections including several episodes of bacterial bloodstream infections. A severe immunoglobulin deficiency was diagnosed and the patient was treated with intravenous immunoglobulin G substitutions. The patient also received long-term treatment with corticosteroids. Elevated aminotransferases were noticed from the end of 2015. In September 2017, he was admitted to our intensive care unit with a septic shock caused by an abscess in the left axilla. During recovery from the sepsis, jaundice was noticed and HEV RNA was detected in his blood (413 mio copies/mL) and feces. RBV therapy was initiated at a reduced dose of 200 mg/d due to his impaired renal function and was subsequently reduced to 200 mg three times per week. In the further course, RBV treatment was increased to a dose of 200–400 mg/d but had to be terminated after 4.5 months due to severe anemia and leukopenia. HEV replication remained continuously detectable throughout the complete RBV treatment period, confirming non-response in this patient.

#### 3.1.4. Case 4

A 68-year old male patient had been diagnosed with marginal zone lymphoma in 2011. He had received chemotherapy (R-CHOP) and autologous hematopoietic stem cell transplantation in 2012. After recurrence of his lymphoma in 2015 he had been treated with ibrutinib from 2015 to 2017 followed by bendamustine and RTX from January to October 2018. He had increased aminotransferases in March 2019, and HEV RNA was measured at 43 mio copies/mL in June 2019, prompting the initiation of treatment with ribavirin at a dose of 600 mg/d, which led to HEV clearance in July 2019. RBV treatment was terminated and the patient showed no HEV replication in October 2019. In November 2019, a viral relapse was detected and retreatment with 600 mg RBV/d was initiated.

#### 3.1.5. Case 5

A 31-year old male patient underwent kidney transplantation due to hypoplastic kidneys in 1997. He developed a severe post-transplant lymphoma (PTLD) requiring chemotherapy (five cycles of RTX and high-dose cytarabine followed by another cycle of RTX monotherapy) in April 2013. In 2015, he experienced terminal renal graft failure leading to the discontinuation of post-transplant immunosuppression consisting of corticosteroids and mycophenolate mofetil. He suffered from immunoglobin deficiency treated with intravenous immunoglobin G substitutions. In May 2019, an abdominal ultrasound revealed a compensated liver cirrhosis. Extended laboratory testing revealed HEV infection which was also detectable in retained blood samples, confirming the diagnosis of a chronic hepatitis E. RBV treatment was initiated (dose of 200 mg/d), but had to be discontinued shortly after due to severe thrombocytopenia and the reduced general condition of the patient.

Table 2 summarizes the underlying disease, immunosuppressive medication, time from last administration of RTX to HEV diagnosis, laboratory results, and HEV serology at the time of HEV diagnosis of the present patient cohort.

Figure 1 shows the clinical course of HEV infection in our five patients, depicting the HEV load under the respective RBV treatment regimens and the development of HEV mutations associated with RBV resistance determined by Sanger sequencing. As depicted in Table 3, patient #1 exhibited hypogammaglobulinemia with decreased levels of IgG, IgA, and IgM. While expressing normal levels of CD19+ cells, class-switched memory B cells (CD19+ IgD-IgM-CD27+) were clearly diminished (0.2% of CD19+). In addition, patient #1 showed a profound T- and CD4-lymphopenia (190/µL absolute counts) and a low percentage of naïve CD4+ cells. In patient #2, no CD19+ B cells were detectable more than 12 months after the last RTX treatment. In agreement with the diagnosis of Waldenstrom’s disease, IgM levels were increased, while levels of IgG and IgA were reduced. Expressing only a mild T-lymphopenia, patient #2 also showed a relevant reduction in CD4+ cells (180/µL absolute cell count) and low naïve CD4+ cells. Patient #3 exhibited lower IgG trough levels and a complete reduction of IgA and IgM under immunoglobulin replacement therapy. No CD19+ B cells were detectable. Moreover, patient #3 also showed CD4-lymphopenia (100/µL absolute cell count) with low naïve CD4 cells. Patient #4 revealed hypogammaglobulinemia with reduced levels of IgG, IgA, and IgM with completely diminished CD19+ cells. In addition, patient #4 showed profound T- and CD4-lymphopenia (180/µL absolute counts) including a low percentage of naïve CD4+ cells (1%). According to his clinical files, patient #5 had normal immunoglobulin levels after kidney transplantation and before the initiation of PTLD chemotherapy. After chemotherapy the patient expressed constant hypogammaglobulinemia with IgG levels as low as 0.75 g/L, IgA levels of 0.2 and no detectable IgM levels. Finally, patient #5 also showed complete B-lymphopenia, NK-lymphopenia, and a severe T-lymphopenia with an absolute CD4 count of 50/µL.

In summary, all patients exhibited relevant hypogammaglobulinemia and T-lymphopenia following RTX-containing treatment regimens in the past.

HEV subgenotypes and potential RBV treatment failure-associated mutations are listed in Table 4.

## 4. Discussion

Persistent HEV infections in immunocompromised patients following RTX-containing treatment regimens have recently emerged [9]. Here we report the largest patient cohort with chronic hepatitis E in this special risk population. In line with previous observations in RTX-treated patients [13,15,17], none of our patients exhibited anti-HEV-IgM-antibodies. This observation adds further evidence to the obvious limitation of serological screening approaches as suggested by current HEV management guidelines [2] and highlights the need for early molecular HEV screening and monitoring in RTX-treated patients with persistently elevated aminotransferases. Moreover, the limited treatment response in our cohort was in line with previous observations suggesting that persistent HEV infection in RTX-treated patients might be difficult to treat by applying standard RBV treatment regimens, since only one of our five patients achieved SVR. On a molecular level, treatment failure to RBV was accompanied by the emergence of RBV treatment failure-associated HEV mutations in our patients (Table 4). These mutations presumably interact in a complex manner resulting in RBV treatment failure. The G1634R mutation is well described and appeared in our cohort in two patients who failed to achieve SVR and in patient #2 who could only clear HEV after a second application of RBV. Another mutation suspected to be relevant in the mechanism of RBV treatment failure is the K1383N mutation which appeared in two patients in our case series that both experienced treatment failure. In addition, other previously described RBV-treatment failure mutations occurred in our patient cohort without a particular accumulation and some mutations previously described could not be detected in any patient. Notably patient #5, who could only be treated with RBV for a very short period of time due to his clinical condition, did not show any of the previously described mutations. Taken together, our findings appear to add further evidence for the potential impact of HEV mutations on the outcome of standard ribavirin therapy, which has been controversially discussed in the past [24,25].

Furthermore, there are a number of relevant observations in the present cohort which need to be addressed. First, the time period between last administration of RTX and the establishment of chronic HEV infection ranged up to 110 months in our cohort, which was considerably longer compared to previous reports that described HEV persistence either while under ongoing RTX treatment or after a maximum time period of 12 months following the last RTX application [15]. Second, Alnuaimi et al. postulated that bendamustine might be the main risk factor in combination with RTX predisposing for the acquisition of chronic hepatitis E [15]. However, our patients #1, #3, and #5 did not receive bendamustine arguing against bendamustine as the main risk factor for HEV chronification. Third, we analyzed the immunological characteristics of our cohort to unravel potential mechanisms underlying RTX-induced HEV persistence, which revealed two major findings. All patients presented with a relevant hypogammaglobulinemia, which required intravenous immunoglobulin replacement therapy in two patients. A detailed analysis of B cell subsets showed severely reduced class-switched memory B cells (CD19+IgD-IgM-CD27+) in one patient, while the absolute B-lymphopenia did not allow a detailed quantification of B cell subsets in the other four patients. In addition, all patients showed a relevant CD4-lymphopenia (all patients < 200/µL) with low levels of naïve CD4 cells (< 10% in all patients). Of note, while hypogammaglobulinemia is an expected and known adverse event following RTX treatment, it has been reported that RTX treatment can induce CD4-lymphopenia [26,27,28].

It remains to be determined whether the hypogammaglobulinemia, the CD4-lymphopenia or the combination of both constitute the major risk factor for HEV persistence following RTX-containing treatment regimens. There is only limited data on the prevalence of acute or chronic HEV infection in patients with hypogammaglobulinemia. Mohamed et al. did not find HEV RNA in 27 patients with primary antibody deficiency and persistently elevated liver enzymes [29]. In addition, increased rates of acute or chronic HEV infection have not been observed in patients with relevant primary antibody deficiency such as common variable immunodeficiency disorders (CVID) [30,31]. With respect to a possible role of T-lymphocytes, particularly in HIV-positive patients, CD4-lymphopenia was found to correlate with increased seroprevalence of anti HEV IgG [10]. A previous treatment with RTX is possibly the common denominator for both observed immunological abnormalities. In patients suffering from rheumatoid arthritis treated with RTX, a reduction of CD4-cells has been described [24,25,26]. Although the long-term duration of CD4-lymphopenia following RTX therapy is not known, RTX-related CD4-lymphopenia in our patients cannot be ruled out. However, due to missing data prior to the onset of RTX treatment and prior to the detection of hepatitis E, we cannot exclude that the observed CD4-lymphopenia resulted as a consequence of the infection rather than the previous RTX therapy or possibly already existed even before initiating RTX treatment. The elements of immune response requirements for efficient HEV clearance remain poorly understood. Specific IgG-antibodies to open reading frame (ORF) 1 and ORF2 and CD4+ and CD8+ effector memory T cells have been associated with control of HEV infection and appear to play a central role in the development of long lasting protection for re-infection with HEV [32,33]. The low CD4+ T-cell counts observed in our cohort might therefore be associated with treatment failure. Of note, T cell-based immunotherapies have recently been discussed as promising novel treatment approaches in chronic hepatitis E [34].

Current treatment guidelines recommend pre-therapeutic testing for hepatitis B virus prior to the initiation of RTX-containing treatment regimens [18]. It remains to be determined whether these recommendations should be extended to HEV testing. Our data do not allow to distinguish between a potential HEV reactivation, which has been controversially discussed in the past [35,36], or de novo infections e.g., by treatment-associated transfusions, which appear to represent a relevant risk for HEV infection for immunosuppressed patients [37].

## 5. Conclusions

Considering the obvious risk of difficult-to-treat HEV persistence in HEV seropositive patients, the administration of RTX should be considered carefully by attending physicians in case of alternate effective treatment options. In addition, potential risk factors for HEV transmission such as consumption of potentially contaminated meat products or blood product administration should be counselled carefully in these patients. In the case of suspected hepatitis, molecular HEV testing appears to be mandatory. In summary, here we report further evidence to prior anecdotal observations suggesting that RTX-containing treatment regimens may imply a long-term increased risk for difficult-to-treat chronic hepatitis E.

## Figures and Tables

**Figure 1 ijerph-17-00341-f001:**
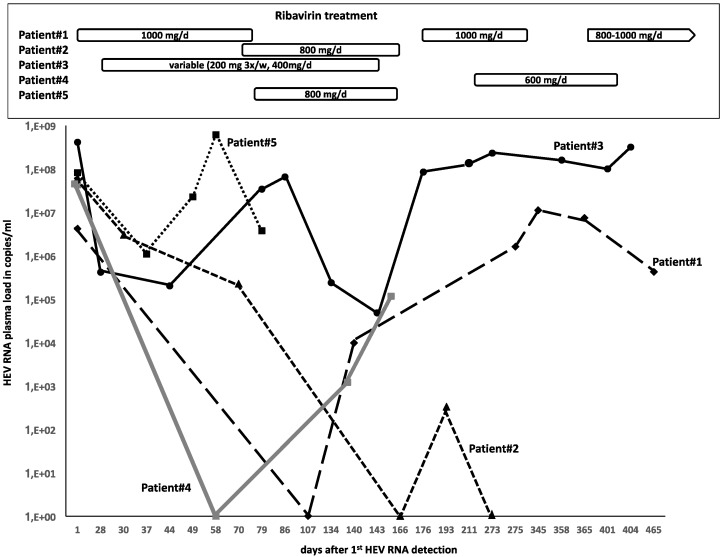
HEV load under ribavirin (RBV) treatment.

**Table 1 ijerph-17-00341-t001:** Primers used for amplification of the RNA-dependent RNA-polymerase (*RdRp*) region and sequencing. HEV: hepatitis E virus.

Primer Name	Sequence (5′–3′)
HEV-247_f	GCHAGGGGGCTYATYCAATC
HEV-128_r	CGGGAYACACGGGTGTTRGTG
HEV-248_r	AACAGCAACARAAYAGCCCT
HEV-27_r	TCRCCAGAGTGYTTCTTCC
HEV-38_f	GAGGCYATGGTSGAGAARG
HEV-39_r	GCCATGTTCCAGACRGTRTTCC
HEV-165_f	TGGAAYACYGTYTGGAAYATGGC
HEV-166_r	CATGTTATTCATTCYAMCCKYTG

**Table 2 ijerph-17-00341-t002:** Underlying disease, immunosuppressive medication, time from last administration of rituximab (RTX) to HEV diagnosis, laboratory results, and HEV serology at the time of HEV diagnosis.

Gender	Age	Underlying Disease	Immunosuppressant	Time Period from Last Administration of RTX to HEV Diagnosis	Anti-HEV IgG	Anti-HEV IgM	ALT (U/L)	AST (U/L)	Total Bilirubin (mg/dl)	GGT (U/L)	INR
Male	69	Non-Hodgkin’s lymphoma	R-CHOP, RTX mono	110 months	Negative	Negative	68	148	0.55	242	0.96
Male	64	Waldenstrom macroglobulinemia	Bendamustine plus RTX	13 months	Positive	Negative	201	103	0.88	289	0.86
Male	22	Atypical hemolytic-uremic syndrome and immune thrombocytopenia	Eculizumab, RTX, corticosteroids	28 months	Negative	Negative	55	111	3.08	1010	1.21
Male	68	Marginal zone lymphoma	R-CHOP, autologous hematopoietic stem cell transplantation, ibrutinib, bendamustine plus RTX	5 months	Negative	Negative	180	88	0.68	180	0.9
Male	31	Post-transplant lymphoma after kidney transplantation	RTX plus high-dose cytarabine RTX mono mycophenolate corticosteroids, 5 mg prednisolone mono from 2015 on	73 months	Negative	Negative	26	32	0.88	368	1.17

Reference ranges: ALT < 35 U/L, AST < 35 U/L, Total bilirubin < 1.20 mg/dL, GGT 8–61 U/L; Abbreviations: ALT—alanine aminotransferase, AST—aspartate aminotransferase, GGT—gamma glutamyl transferase, INR—international normalized ratio, R-CHOP—rituximab, cyclophosphamide, doxorubicin, vincristine, prednisolone.

**Table 3 ijerph-17-00341-t003:** Immunological parameters of the patient cohort.

	Patient #1	Patient #2	Patient #3	Patient #4	Patient #5
IgG (7–16 g/L)	4.12	3.04	4.69	5.44	2.14
IgA (0.7–4.00 g/L)	0.36	0.29	<0.1	0.36	0.18
IgM (0.4–2.30 g/L)	0.07	12.46	<0.05	0.07	0.07
Specific anti-pneumococcal-IgG (10–191.20 mg/L)	41.81	10.12	unknown	115.11	unknown
CD19+ (0.1–0.4/nl)	0.18	0.00	0.00	0.00	0.00
CD19+IgD+CD27- (42.6–82.3%)	93.7	n.a.	n.a.	n.a.	n.a.
CD19+IgD+IgM+CD27+ (7.4–32.5%)	3.8	n.a.	n.a.	n.a.	n.a.
CD19+IgD-IgM-CD27+ (6.5–29.1%)	0.2	n.a.	n.a.	n.a.	n.a.
CD19+CD21lowCD38low (0.9–7.6%)	0.6	n.a.	n.a.	n.a.	n.a.
CD19+CD21lowCD38++IgM+ (0.6–3.4%)	1.3	n.a.	n.a.	n.a.	n.a.
CD19+CD21lowCD38++IgM- (0.4–3.6%)	0.1	n.a.	n.a.	n.a.	n.a.
CD3+ (0.9–2.2/nl)	0.29	0.89	0.42	0.52	0.08
CD3+CD4+ (0.5–1.2/nl)	0.19	0.18	0.1	0.18	0.05
CD3+CD8+ (0.3–0.8/nl)	0.09	0.67	0.3	0.31	0.03
CD3+CD4+CD45RA+ (>15%)	9	4	8	1	n.d.

Abbreviations: n.a.—not assessable due to B-lymphopenia; n.d.—not determined.

**Table 4 ijerph-17-00341-t004:** HEV subgenotypes and potential RBV treatment failure-associated mutations.

	Patient #1	Patient #2	Patient #3	Patient #4	Patient #5
Sample used for analysis (days after first positive PCR)	345	70	176	1	27
HEV subgenotype	3c	3c	3c	3 (could not be assigned)	3c
Y1320H	wt	wt	wt	wt	wt
K1383N	mut	wt	mut	wt	wt
D1384G	mut	wt	wt	wt	wt
K1398R	wt	wt	wt	wt	wt
V1479I	wt	wt	wt	mut	wt
Y1587F	wt	wt	mut/wt	wt	wt
G1634R	mut	mut	mut/wt	wt	wt

Abbreviations: wt—wild-type; mt—mutant. Amino acid numbering throughout the manuscript is according to NCBI reference sequence NP_056779 as proposed by Debing et al. [20,21].

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
