# Peer review of "Rituximab-Containing Treatment Regimens May Imply a Long-Term Risk for Difficult-To-Treat Chronic Hepatitis E"

_ijerph, 2020, doi:10.3390/ijerph17010341_

Round 1

Reviewer 1 Report

It has been recognized that HEV persistence infection often occurs in immunosuppressed patient populations or HIV infected patients with low CD4+ T cell counts. Recently, the anti-CD20 mAb rituximab has been found to be involved in chronicity of HEV infection in patients with haematological malignancies. This work reported 5 chronic hepatitis E patients after prior rituximab therapy. The immunological characteristics of the 5 patients has been determined and the ribavirin treatment failure-associated mutations of viral genome has been analyzed. These cases imply a risk of persistency of HEV infection in people receiving rituximab-containing treatment.

Comments:

In figure 1, the five patients line type is hard to tell. Authors may replace different line types with different hollow symbol to make it easy to read. Line 58-61, the authors should introduce the researches or cases of reference 8, 11-16 in more detail. In table 2, the abbreviations like ALT, AST, GGT, INR, should be noted below the table.

Reviewer 2 Report

This is a mini-cohort study of five patients who developed chronic hepatitis E virus infection after rituximab (RTX) therapy. Due to the treatment, all five patients developed hypogammaglobulinemia and T-lymphophenia, potentially emphasizing the importance of the T-cell response in limiting HEV infection.  Interestingly, three out of five patients who were treated with ribavirin, failed to reach sustained virological response and developed mutations that were previously associated with ribavirin failure. Based on their observations, the authors recommend molecular HEV testing in case of suspected hepatitis and carefully monitoring treatment response.

The observations are interesting and the proposed recommendations should be implemented in clinical procedures. Yet, the study remains anectodical and firm conclusions are difficult to draw.

In general, I would recommend proof-reading the references and making sure that previous reports are properly acknowledged and cited. For example, and among others, the study showing an in vitro inhibitory effect of sofobuvir on HEV infection is missing.

In the introduction, more information on rituximab treatment and reported effects would be helpful for the reader. Also, how do the findings on this cohort compare to previous case reports on chronic HEV infection in RTX-treated patients?

Reviewer 3 Report

The authors have presented 5 cases that indicate that rituximab-containing treatment regimens might imply a relevant risk for persistent HEV infection even years after the last rituximab application.

To my opinion it is not clear that the only factor of immunosuppression is linked to rituximab: all the patients suffered from hematological malignancies or are solid-organ transplant recipient. Thus, their conclusion should be more nuanced.

Moreover, they have stated in the discussion that "there is a need for early molecular HEV screening and monitoring in RTX-treated patients with persistently elevated aminotransferases." However, the ALT or AST values of the five patients are not presented. I also suspect that the acute phase of the infection was not diagnosed and this acute phase may have occured shortly after RTX introduction.

They  have described the mutations in the polymerase that can appear after ribavirin failure. The real impact on the ribavirin therapy is largely debated and should be discussed regarding the study by Lhomme et al AAC 2016; or the vey recent publication by Kamar et al CID 2019. It is surprising to present a table with results only in teh discussion section.
